# The Microstructure, Rheological Characteristics, and Digestibility Properties of Binary or Ternary Mixture Systems of Gelatinized Potato Starch/Milk Protein/Soybean Oil during the In Vitro Digestion Process

**DOI:** 10.3390/foods12132451

**Published:** 2023-06-22

**Authors:** Yufang Guan, Watcharaporn Toommuangpak, Guohua Zhao, Siwatt Thaiudom

**Affiliations:** 1School of Food Technology, Institute of Agricultural Technology, Suranaree University of Technology, Nakhon Ratchasima 30000, Thailand; 2The Food Processing Research Institute of Guizhou Province, Guizhou Academy of Agricultural Sciences, Potato Engineering Research Center of Guizhou Province, Guizhou Key Laboratory of Agricultural Biotechnology, Guiyang 550006, China; 3College of Food Science, Southwest University, Chongqing 400715, China

**Keywords:** potato starch-based foods, digestibility, rheology, microstructure, milk protein

## Abstract

The in vitro digestibility of potato starch-based foods interacting with milk protein and soybean oil was investigated. Microstructures and rheological changes upon digestion were determined. The results showed that the addition of milk proteins (casein and whey protein) promoted gelatinized potato starch digestion, while soybean oil slowed down gelatinized potato starch digestion. A mixture of soybean oil and milk protein promoted the digestion of milk protein, while a mixture of gelatinized potato starch and milk protein inhibited the digestion of milk protein. The mixture of milk protein and/or gelatinized potato starch with soybean oil promoted the release of free fatty acids in soybean oil. The highest release rate of free fatty acids was attained by a mix of milk protein and soybean oil. The mixed samples were digested and observed with a confocal laser scanning microscope. The viscosity of the digestates was determined by a rheometer. Overall, the results demonstrated that the addition of milk protein and soybean oil had an effect on the in vitro digestibility of gelatinized potato starch and its microstructure.

## 1. Introduction 

Starch is a major food source for humans, accounting for about 70 percent of the calories in the human diet. Starch-based foods are products containing high amounts of starch, but these foods often also contain a certain number of proteins and fats. There are many studies on the binary interaction between starch and proteins or lipids [1,2,3,4]. Mixing starch with proteins or lipids affects the digestibility of starch [5,6,7]. Amylose is known to form a single spiral complex with lipids [8], and it combines with lipids and other similar compounds recognized as being resistant to α-amylase [9]. For the interaction of starch and protein, the addition of protein to starch might hinder or promote starch digestion [10,11]. However, generally, these three ingredients are often present in general food at the same time, and their interactions further affect certain physiological responses in humans, such as postprandial blood sugar levels [12,13].

The influence of proteins and lipids on the starch digestibility of kodo millet flour and rice flour was investigated [14]. The study results confirmed that proteins and lipids inhibit starch digestion. Proteins and lipids might slow down the hydrolysis of starch by inhibiting starch swelling, covering starch granules, and restricting the digestive enzymes to entry into the starch molecules. Chen et al. (2017) [15] studied the effect of the addition of soy protein and corn oil to corn starch on starch digestibility and found that the complex reduced the rapidly digestible starch (RDS) and increased the slowly digestible starch (SDS) and resistant starches (RS). In addition, the impact of protein from soy on the digestibility of that ternary mixture was greater than that of corn oil, and the physical barrier of corn oil, a protein-starch matrix and an amylose-lipid complex, provided resistance to starch digestion [15]. Thus, the degree of unsaturation of fatty acids and chain length affected the in vitro digestibility of starch-protein-fatty acid complexes [16]. 

Starch-based foods such as bread, mashed potatoes, and noodles contain not only starch but also protein and lipids. Processing conditions and the composition of raw materials could affect the microstructure, rheological characteristics, and digestibility of these foods. Food processing could influence the glycemic index (GI) of potato starch products [17], while the rheological properties of starch-based emulsions containing whey protein and oil could affect fat digestibility and fat release from the potato network [13]. The effect of rheological characteristics on digestibility was also found in a work by Jin et al. [18], in which the higher viscosity of peanut butter could retard protein hydrolysis and lipid digestion. In vitro and in vivo studies of several cultivars of potatoes demonstrated that cooked potatoes exhibited the fastest starch digestion rate and absorption rate in humans, making the levels of postprandial blood glucose higher [19]. Highly digestible carbohydrate-rich foods lost their popularity due to their effect on postprandial blood glucose level elevation when consumed, which can cause physiological complications associated with obesity and diabetes. Even though there is considerable research determining the interactions among starch, protein, and lipids and their effects on starch digestibility [20,21], a need for a greater understanding of the effects of adding specific milk proteins (MP) such as casein (CA), whey protein (WP), and soybean oil (SBO) to cooked potatoes still needs to be investigated, as well as the correlation between the rheological properties and digestibility of such ingredients and their interaction. 

Thus, this study focused on the effect, in terms of interactions, of adding CA or WP and SBO to gelatinized potato starch (GPS) on GPS properties: changes in GPS microstructure, GPS rheology, and GPS digestibility. In addition, the effect of CA, WP, and SBO on starch hydrolysis in GPS was studied, along with its mechanism. New findings result from this study, which could benefit food manufacturing of potato starch in terms of applying such results to design a more suitable process and product in the future.

## 2. Materials and Methods

### 2.1. Materials and Reagents 

Potato starch (PS) was provided by Bangkok Inter Food Co., Ltd. (Bangkok, Thailand). CA powder with 82% (*w*/*w*) total protein was purchased from Vicchi Enterprise Co., Ltd. (Bangkok, Thailand). WP with 80% total protein was received from Shanghai Yuanye Biological Technology Co., Ltd. (Shanghai, China). SBO (Thanakorn Vegetable Oil Products Co., Ltd., Samut Prakan, Thailand) used in this study was bought from a convenience store in Nakhon Ratchasima province, Thailand. α-Amylase from porcine pancreas (A-3176; Type VI-B), pepsin from porcine gastric mucosa (P7000), and pancreatin from porcine pancreas (P7545, activity 4 × USP) were bought from Sigma-Aldrich Chemical Co. (St. Louis, MO, USA). Megazyme International Ireland Ltd. (Co. Wicklow, Ireland) provided the assay kit of glucose oxidase-peroxidase (GOPOD), the assay kit of total starch, and amyloglucosidase (3200 U/mL). All chemicals used in this study were of analytical grade. 

### 2.2. Proximate Analysis

The fat, protein, ash, and moisture content of the raw materials were measured using official AOAC methods (AOAC 2000). Fatty acids (FAs) of the SBO were determined according to the method of AOAC 969.33 (2000) by Gas chromatography (7890 GC system, Agilent Technologies Inc., Santa Clara, CA, USA). The initial amount of starch in the samples was evaluated using an analytical kit for total starch. 

### 2.3. Sample Preparation

The potato starch-based foods were prepared according to the previous research methodology of Guan et al. [22]. Briefly, PS slurry (6.0 g in 40 g deionized water) was gelatinized in a water bath at 95 °C for 30 min, while MP was dissolved in 95 °C deionized water. SBO (0.9 g) was also heated in a 95 °C water bath for 30 min. The mixtures were prepared as shown in Figure 1. Thus, four primary systems (GPS, CA, WP, and SBO), five binary systems (GPS/CA, GPS/WP, GPS/SBO, CA/SBO, and WP/SBO), and two ternary systems (GPS/CA/SBO and GPS/WP/SBO) were set up for the experiment. In all systems, PS, WP, and SBO were present at constant concentrations of 10.0 g/100 g wb, 1.0 g/100 g wb, and 1.5 g/100 g wb, respectively. The samples were first frozen at −80 °C in a freezer (Haier, Qingdao, China) and then dried at −20 °C in a vacuum freeze-dryer (Seientz-10ND, Ningbo Xinzhi Biotechnology Co., Ltd., Ningbo, China) for the analysis that followed. The freeze-dried samples were ground to pass through a 60-mesh sieve prior to the measurement. Until the next experiment, the ground samples were hermetically sealed in a plastic bag and kept at room temperature. 

### 2.4. In Vitro Starch Digestion

A rapid in vitro starch digestion assay following the methods of Sopade and Gidley was used [23]. Test samples (2.5 g) were treated with 1 mL of artificial saliva, which contained porcine α-amylase (250 U/mL of carbonate buffer, pH 7) for 15–20 s. Thereafter, 5 mL of pepsin (1 mL/mL of 0.02 M aq. HCl) was added to the samples, and then the samples were incubated for 30 min in a water bath (37 °C) rotating at 85 rpm. The digested samples were neutralized with 5 mL of 0.02 M aq NaOH. Then, their pH was adjusted to 6 using 25 mL of 2 M sodium acetate buffer. Finally, 5 mL of mixed solution was prepared from amyloglucosidase (28 U/mL of acetate buffer) and pancreatin (2 mg/mL of acetate buffer), which were added to the digestion tube containing the digested samples. These samples were incubated at 37 °C for 4 h.

Aliquots (0.5 mL) were withdrawn at 0, 10, 20, 30, 45, 60, 90, 120, 150, 180, 210, and 240 min of digestion during the incubation in the intestinal stage, followed by mixing with 95% ethanol (3 mL). The aliquots were then analyzed for glucose using the GOPOD-assay kit. The starch hydrolysis was calculated with the following equation [24]:(1)$$\%S_{H}=\frac{S_{h}}{S_{i}}=0.9\times \frac{G_{p}}{S_{i}}$$
where % *S_H_* was the percentage of starch hydrolysis (total), *S_h_* was the amount of starch hydrolyzed, *S_i_* was the initial amount of starch (g), and G_p_ was the amount of glucose produced (g). Generally, 0.9 was used as the conversion factor from starch to glucose, which was calculated from the molecular weight of starch monomer/molecular weight of glucose (162/180 = 0.9). 

### 2.5. Estimation of Glycemic Index (GI)

The digestogram showing digested samples at a specific time period was modeled using Equation (2) [23].
(2)$$D_{t}=D_{\infty -0}\;\left(1-\exp\left[–Kt\right]\right)$$
where *D_t_* (g/100 g dry starch) was the digested starch at time *t, D_0_* was the digested starch at time *t* = 0, *D∞* was the digestion at infinite time (*D_0_* + *D_∞−0_*), and K was the rate constant (min^−1^). *D_∞−0_* was estimated from t = 0–240 min.

In order to calculate the estimated *GIs* of the samples, the areas under the digestograms (*AUCexp*) were computed with Equation (3) [23]:(3)$$AUC_{exp}=[D_{\infty }t+\frac{D_{\infty -0}}{K}exp\;(-K{\left.t)\right]}_{t_{1}}^{t_{2}}\;$$

Estimated GI values were determined following the method of Goñi with some modifications [25]. The estimated GI was achieved by using single-point measurements of starch digestion at 90 min. The hydrolysis index (*HI*) of each test sample was determined by dividing the area under the digestogram of the sample with the area under the digestogram of fresh white bread, which was around 17,000 min g/100 g dry test sample (from 0 to 240 min) in this study. Using the parameters of the modified first-order kinetic model for the test samples and fresh white bread, GI (average) was estimated (GIAVG) for each sample and was calculated using Equation (4) [26]:(4)$$Estimated\;GI=\left[\frac{\left(\left(39.71+0.803H_{90}\right)+\left(39.51+0.573HI\right)\right)}{2}\right]$$

### 2.6. Protein Digestion

The in vitro protein digestibility of the test samples was carried out following the method of Srigiripura and Kotebagilu et al. [27]. All test samples except GPS were taken in quantities equivalent to 100 mg of protein (about 10.0 g) for analysis. The protein content of these samples was determined by the Kjeldahl method. The samples (50 mL) were put into the centrifuge tubes mixed with 15 mL of 0.1 N HCl, containing 1.5 mg of pepsin. Then, these samples were incubated at 37 °C for 3 h. Then, 1.5 mL of 0.5 N NaOH and 7.5 mL of 0.2 M phosphate buffer, containing 4 mg of pancreatin, were added to the samples before incubation at 37 °C for 24 h. Thereafter, 10% trichloroacetic acid was added to stop the incubation, and the mixture was allowed to stand for 2 h. Afterwards, the samples were centrifuged at 4000 rpm for 15 min, while the Kjeldahl method was used to analyze the protein content present in the supernatant. The protein digestibility was computed using the following formula (5) [27]:(5)$$Protein\;Digestibility\left(\%\right)=\frac{Protein\;content\;in\;the\;supernatant}{Total\;protein\;content\;of\;the\;samples}\times 100$$

### 2.7. Oil Digestion 

The intestinal digestion model was modified slightly from the methodologies of Hu et al., Qin et al., and Wan et al. [28,29,30]. Twenty g of SBO, GPS/SBO, CA/SBO, WP/SBO, GPS/CA/SBO, and GPS/WP/SBO were weighed in a glass beaker and placed into a water bath at 37 °C for 10 min. The sample pH was adjusted to 7.0 using NaOH (0.5 N) or HCl (0.1 N) solutions. Then, 3.5 mL of preheated bile extract solution and 1.5 mL of mineral ion solution were added to the sample under continuous stirring, followed by pH readjustment of the sample back to pH 7.0. Two and a half milliliters of freshly prepared porcine pancreatin suspension (60 mg pancreatin powder dispersed in 5 mM phosphate buffer, pH 7, 37 °C) were added to the sample in order to initialize the titration. During in vitro intestinal digestion, the pH was monitored using a pH meter (SevenCompact™ S210-S, Mettler Toledo International Trade Co., Ltd., Shanghai, China) at every 2 min until 120 min after digestion and maintained at 7.000 ± 0.005 by adding 0.5 M NaOH through a burette. A recording of volumes of 1 M NaOH used to neutralize the FFA indicated the FFA released, which was produced by the triglycerides (supposing that two FFAs were released per triglyceride), following Equation (6) [29,31]:(6)$$FFA\%=100\times \frac{\left(V_{NaOH\;for\;sample}-V_{NaOH\;for\;blank}\right)\times C_{NaOH}\times MW_{lipid}}{2W_{lipid}}$$

Here, *V_NaOH for sample_* was the volume of NaOH (L) titrated into the reaction vessel to neutralize the FFAs released; *V_NaOH for blank_* was the volume of NaOH (L) titrated into the reaction vessel to neutralize the FFAs released in the absence of oil. *C_NaOH_* was the concentration of NaOH (0.1 M); *W_lipid_* was the initial mass of SBO (g) in the intestinal phase; and *MW_lipid_* was the average molecular weight of corn oil (872 g mol^−1^). 

### 2.8. Microstructure Analysis 

Samples prepared in 2.3 were taken for microstructure analysis during the simulated digestion process, in which these samples were digested in the in vitro stomach stage for 15 min and in the in vitro small intestine stage for 5 min. The changes in these samples were observed with confocal laser scanning microscopy (CLSM) using the fluorescent mode (Nikon A1R, Nikon Crop., Tokyo, Japan) according to the method of Thaiudom and Pracham [32]. APTS (8-amino-1,3,6-pyrenetrisulfonic acid) and Nile Red in distilled water were used to dye GPS and SBO, respectively. Fluorescein isothiocyanate isomer I (FITC) in acetone was used to dye both GPS and CA. APTS developed a blue color, while FITC and Nile red developed green and red colors, respectively. 

### 2.9. Rheology 

The viscosity of GPS, binary, and ternary systems during simulated in vitro digestibility in 2.4 was investigated following the method of Bordoloi, Singh, and Kaur [33]. Time sweep experiments were conducted with a dynamic rheometer (AR-G2 Rheometer, TA Instruments, New Castle, DE, USA) equipped with a peltier cylinder system. The measurements were achieved using a cup and vaned rotor geometry. Two and a half grams of the sample were weighed into a 50-mL centrifuge tube, and then 1 mL of artificial saliva containing porcine α-amylase (250 U/mL of carbonate buffer, pH 7) was added to the samples and mixed for 20 s. Then, 20 mL of pepsin solution (1 mL/mL of 0.005 M HCl, pH 2) was added. The mixtures of samples and enzyme solution were immediately loaded into the rheometer cup. The experiment was conducted at 37 °C using a multi-step flow procedure following Qin et al. [28]. Control potato starch paste (Control-GPS) was prepared in the same way without the addition of enzymes and set as the control.

### 2.10. Statistical Analysis

Measurements for all the experiments were performed at least in duplicate. The results were exhibited as the mean ± standard deviation. Statistical analyses were determined using SPSS 23.0 (SPSS Statistical Software, Inc., Chicago, IL, USA). An analysis of variance (ANOVA) was carried out to determine differences. A *p*-value less than 0.05 was considered as a significant difference.

## 3. Results

### 3.1. Chemical Composition

Table 1 shows the chemical composition of PS, CA, WP, and SBO. The main component of PS was 76.77% starch containing amylose about 31.14% *w*/*w*. Protein was the main component of CA and WP, and the protein content was 81.25 and 78.67% *w*/*w*, respectively. The main component of SBO was 99.9% *w*/*w* fat. 

### 3.2. In Vitro Starch Digestibility 

The effect of MP addition (CA or WP) and/or SBO on the starch digestibility of GPS is shown in Figure 2. In the simulated oral phase (first stage, 15–20 s), α-amylase hydrolysis of a small number of PS occurred, but glucose was rarely produced. In the gastric phase (second stage, 30 min), no glucose was released from those samples. This is due to the low pH environment at this stage, which leads to α-amylase enzyme inactivation [33]. The detection of glucose content in the whole in vitro digestion process began at the end of the gastric phase. However, a small amount of glucose was found at the end of the gastric phase. This was attributed to starch hydrolysis in the simulated oral cavity [34].

When pancreatin and amyloglucosidase were added to the test sample, GPS/CA, GSP/WP, GSP/CA/SBO, and GSP/WP/SBO were rapidly digested within the first 30 min. At 30 min, starch digestibility reached 30–40%. The digestion rate of GPS and GPS/SBO during this period was relatively slow, and the digestibility was less than 10% after half an hour. These differences might be due to the composition of the individual test samples [35,36]. The digestion of these compositions occurred with specific enzymes throughout the simulated digestion process. Pepsin could digest MP and change the structures of those samples during digestion in the simulated gastric phase. The overall structure of those samples became looser when the protein was first digested, so when pancreatin and amyloglucosidase were added into the simulated GI system, the digestion rate of those samples then became significantly faster than that of GPS and GPS/SBO. Half an hour later, the starch hydrolysis rate of all samples was slower until the end of the digestion. At 240 min, the digestibility of those samples reached 90–95%. The final digestibility of GPS and GPS-SBO was between 60–65%. Quantitatively, the digested starch obtained in those samples was better than that obtained in GPS, while the digested starch obtained in GPS-SBO had a lower value than that of GPS. Thus, MP played a key role in enhancing the hydrolysis of starch. There were many possible reasons why the presence of MP might increase the enzymatic digestion of starch. Firstly, the decrease in viscosity of GPS after adding MP might enhance enzyme susceptibility and hydrolysis [10], which could be obviously seen in the rheological study part mentioned further. Secondly, the MP in a small practical state in the binary or ternary system could disrupt the GPS gelling 3D network, making it easier for the enzyme to contact the starch (as shown in Figure 3). In contrast, the inhibitory effect of SBO on starch digestion could be attributed to the hydrophobic interaction between starch and oil [21].

### 3.3. Hydrolysis Kinetics and Estimated Glycemic Index

The important factor in the glycemic response was the rate of starch loosening. The in vitro investigations predicted postprandial results with good accuracy. Table 2 shows the in vitro digestion findings, including the equilibrium concentration (C1) at 180 min, the *kinetic constant* (*k*), the *hydrolysis index* (*HI*), and the *estimated glycemic index* (*eGI*). 

The addition of SBO had no significant effect on the kinetic constants of GPS, while the addition of MP to GPS significantly increased the kinetic constants of GPS. The viscosity of the samples that contain PS was measured. It can be seen from the results that the smaller the viscosity, the larger the kinetic constant, and the easier the GPS was digested, the higher the eGI value. When MP was added to GPS, the voided spaces could be seen in the three-dimensional structure of GPS. This assisted α-amylase to penetrate GPS more easily while SBO infiltrated the GPS 3-D structure, which might have led to hydrophobic interaction between GPS and SBO. Thus, GPS was more tightly bound to SBO, and sequentially, the α-amylase rarely digested the starch (Figure 3a,b,d).

### 3.4. Protein Digestion

CA, WP, GPS/CA, GPS/WP, CA/SBO, WP/SBO, GPS/CA/SBO, and GPS/WP/SBO possess protein contents of 0.990%, 0.987%, 0.958%, 0.959%, 0.939%, 0.997%, 0.971%, and 0.934%, respectively. The protein content of all samples was close to 1.0 % and not significantly different among those samples. The protein digestibility of the test samples is shown in Figure 4. For single-phase samples, WP possessed a higher protein digestibility than CA. The results agree with previous research studies [37,38]. This was due to the average diameter of casein micelles being around 120 nm, which was larger than that of WP. The enzyme has more surface area in contact with WP, so it is easier to digest than CA. The protein digestibility of CA/SBO and WP/SBO in binary systems was higher than that of CA and WP, respectively. This was possible since parts of *β*-lactoglobulin and *β*-casein were located at the oil-water interface. They were more easily broken down by pepsin when adsorbed at the oil-water interface than in solution [39,40]. The proteins unfold on the surface of the droplets, improving their accessibility to pepsin [41]. On the contrary, the digestibility of GPS/CA and GPS/WP was lower than that of CA and WP, respectively. The viscosity of GPS/CA and GPS/WP was much larger than that of CA and WP because of the effect of hydrated PS. The high viscosity of GPS hindered the movement of MP and prevented the accession of enzymes to digest MP [42]. For the ternary system, the protein digestibility of GPS/CA/SBO and GPS/WP/SBO was lower than that of GPS/CA and GPS/WP, respectively. This may be because CA and WP could adsorb on the surface of the SBO after it was added to GPS/MP, which made the ternary system more stable. The protein hydrolysis of MP was inhibited in the more stable ternary system, so the protein digestibility of the ternary system is lower than that of the binary system (GPS/CA and GPS/WP).

### 3.5. Free Fatty Acid Release

The fat digestibility of all samples containing SBO was determined. Total FFAs released from the oil phase are shown in Figure 5. When the digestion time increased, the released amounts of FFA in each sample also promptly increased, especially rapidly in the first 30 min, and then flattened out after 60 min. This was because the lipolytic products accumulating at the oil-water interface could inhibit pancreatic lipase from entering the triglyceride core [43]. After 2 h of in vitro intestinal digestion, the FFA release ratio of the samples with different components was different (Figure 5). The total release of FFA from SBO was the lowest, at about 50%. This might be because oil is incompatible with water, so the oil clumps together during digestion, reducing the reaction surface of the oil. The fat digestibility of GPS/SBO was higher than that of oil alone. After the starch granules were gelatinized, the dispersed starch chains reassociated with the cooled gelatinized starch [44]. The SBO dispersed into the gelatinous structure of the starch during cooling. So, the mixing of GPS and SBO could have made SBO less likely to aggregate than SBO in water, so that more could be released into the intestinal fluid [45]. However, in binary systems, CA/SBO and WP/SBO possess a higher total release of FFA. This might be attributed to the emulsification property of CA and WP after homogenization, which increased the surface available to the lipase [41], resulting in a higher release of FFA. In the ternary system, the test sample mixed with GPS made the total release of FFA lower than that in the CA/SBO and WP/SBO. This was because the addition of GPS increased the viscosity of the system, leading to a decrease in the activity of lipase at the interface layer, thereby delaying the release of FFA [46]. However, ternary systems composed of GPS and SBO with different MPs have a higher FFA release than binary systems without MPs. This might be because of the emulsifying properties of CA and WP, which could disperse the oil more in the ternary system than in the binary system of GPS/SBO. So, the enzyme may have entered the oil particles easier in a ternary system than in a binary system.

### 3.6. Microstructural Changes during Digestion

All samples containing GPS and the state of each sample in the process of simulated intestinal conditions were observed by CLSM. The observation results are shown in Figure 3. The blue color of ATP-stained GPS and the green color of FTIC-stained GPS can be clearly seen in the figure. The superposition of the two colors resulted in cyan (as shown in Figure 3a). As shown in the figure, the GPS had a uniform texture with no granular matter, showing that the GPS was completely gelatinized, and the particle structure of the GPS was destroyed after homogenization. The images obtained from the simulated digestion process show that their green coloration was reduced in luminosity when the digestion time increased, just as the blue color of the image became darker when the digestion time reached 35 min. This can be attributed to the decrease in starch concentration upon the addition of enzyme solution in the gastric and intestinal phases [47]. In addition, the GPS was hydrolyzed after the addition of pancreatin and amyloglucosidase.

In the binary system, regarding the GPS/CA and GPS/WP samples, it can be seen from Figure 3 (b-GPS/CA-0, c-GPS/WP-0) that CA and WP with a bright green color were scattered in the 3D network of GPS. This phenomenon is similar to the phase separation of polysaccharides and proteins [32,48,49,50,51,52]. Figure 3 (b-GPS/CA) shows that the concentration of CA was significantly reduced during gastric simulated digestion (15 min, 30 min) due to the addition of pepsin during this process, which caused partial digestion of CA. It also shows (c-GPS/WP) partial digestion of WP after 30 min of digestion in the gastric phase. However, the image of 5-min digestion in the small intestine stage shows voided spaces, which were caused by the phase separation between GPS and MP. 

The observation results of GPS/SBO are shown in Figure 3 (d-GPS/SBO-0). The SBO droplets, strained with Nile red, can be seen infiltrating the spaces in the GPS 3D structure. The overlapping of GPS with SBO is shown by a pink color. This indicates that there was an interaction between SBO and GPS in this sample, which might have formed a starch-lipid complex [21]. During simulated digestion in the gastric phase (15 min and 30 min), the oil droplets, stained by FITC green, also overlapped with Nile red and finally showed a yellow color. After simulated digestion in the small intestine stage for 5 min, the digested sample was diluted with the addition of enzyme solution, and a part of the GPS was digested quickly. Some void spaces can be seen in the image. Pancreatin in digestive juices containing lipases digests oil, and the hydrolysis of oil causes the appearance of cavities [53]. 

Regarding the ternary system, the images of GPS/CA/SBO (Figure 3e) and GPS/WP/SBO (Figure 3f) indicate that CA, WP, and SBO infiltrate the GSP 3D network. Compared with the GPS/SBO binary system, the dispersion of SBO in the ternary system was efficient, and the distribution of oil droplets was more uniform. This is because MP, as the emulsifier, emulsifies, making the ternary system more stable during homogenization [54]. Some other phenomena were also observed from the images of GPS/CA/SBO (Figure 3e) and GPS/WP/SBO (Figure 3e). During the simulated digestion, protein was partially digested in the samples of the ternary system. GPS and SBO were diluted and digested during the intestinal phase (35 min). This observation is consistent with the actual consumption of protein and oil in the human body.

### 3.7. Rheology—Flow Behavior

The viscosity of the test samples containing GPS during the simulation of in vitro digestibility is shown in Figure 6. The viscosity of the test samples depended on the composition and interaction between the MS, SBO, and GPS [55,56,57]. In in vitro digestion, Control-GPS had a considerably higher viscosity than GPS, GPS/CA, GPS/WP, GPS/SBO, GPS/CA/SBO, and GPS/WP/SBO, which were all mixed with different enzymes. In the simulated gastric phase without the addition of enzymes, the viscosity of GPS increased at the first stage of digestion and then slightly decreased before remaining basically unchanged (Figure 6). The viscosity of GPS increased slowly in the simulated stomach stage with the addition of enzymes and then remained basically stable. However, the viscosity of GPS was lower than that of Control-GPS. This was because the GPS was added with porcine α-amylase prior to the simulated stomach stage, resulting in a lower viscosity than Control-GPS. In the gastric stage, the viscosity of GPS/SBO was the highest, followed by GPS. While the binary and/or ternary mixtures with MP showed slightly lower viscosity during digestion in the gastric stage than GPS/SBO and GPS. There are two possible explanations for this result. One is that the viscosity of GPS/CA, GPS/WP, GPS/CA/SBO, and GPS/WP/SBO was lower than that of GPS/SBO and GPS [22]. The other explanation might be that MP in those samples was hydrolyzed by pepsin in the gastric phase [58]. This could have destroyed the structure of MP after hydrolysis and also caused the 3D networks of the GPS in the sample to become loose, thus reducing the viscosity of those samples.

With the addition of the enzymes prior to the simulated digestion in the intestinal stage, the viscosity of the sample in the reaction tube decreased, not only because the test sample was diluted by the addition of the enzyme solution but also because of the rapid hydrolysis of the sample by the addition of enzymes. Compared with Control-GPS, the viscosity of each sample with the added enzyme solution was significantly lower due to the hydrolysis of GPS, SBO, and MP with the addition of specific enzymes. GPS was hydrolyzed by α-amylase and amyloglucosidase to sugar, dextrins, and maltooligosaccharides, which possess a lower viscosity than GPS [59]. SBO was hydrolyzed by lipase to FFA [59,60]. Proteases hydrolyze MP into small molecular polypeptides [61]. So, the viscosity of the samples that were added to the corresponding enzyme solution was significantly lower than Control-GPS. Thus, it seemed to be that the higher the viscosity of the sample, the lower the starch digestibility in these studied binary and ternary systems.

## 4. Conclusions

The addition of MP and/or SBO to GPS changed the digestibility properties of the GPS. SBO inhibited the digestion of GPS, while the addition of MP promoted the digestion of GPS. The mixture of SBO and MP (CA or WP) promoted the digestion of protein, while the mixture of GPS and MP inhibited it. The mixtures of GPS/MP/SBO inhibited protein digestion. However, this inhibition was found to be at a higher rate in the sample of GPS/SBO than in GPS/MP/SBO. A mixture of MP and/or GPS with SBO promoted the release of FFA in SBO, while a mixture of MP with SBO resulted in the largest release of FFA in SBO. CLSM was an effective tool for revealing the changes in the binary and ternary systems during simulated digestion. Mixing SBO with GPS increased the viscosity of GPS, while mixing MP with GPS decreased the viscosity of GPS, resulting in inhibiting and increasing GPS digestibility, respectively. In conclusion, the evidence found in this study could be of use as guidance for food processing and food product development, especially in the field of food ingredient interaction in dairy and starch technology.

## Figures and Tables

**Figure 1 foods-12-02451-f001:**
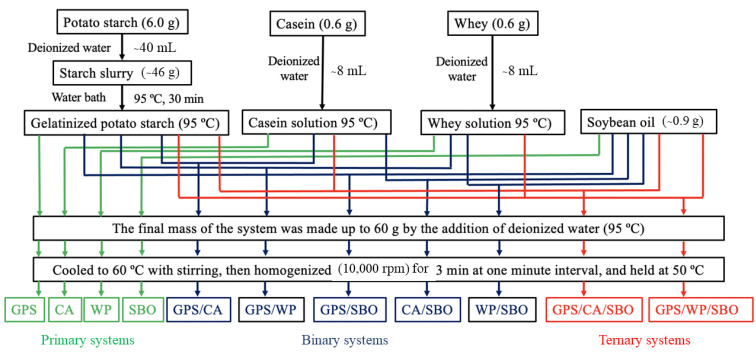
Flow chart of sample preparation with different ingredients to produce different interaction systems.

**Figure 2 foods-12-02451-f002:**
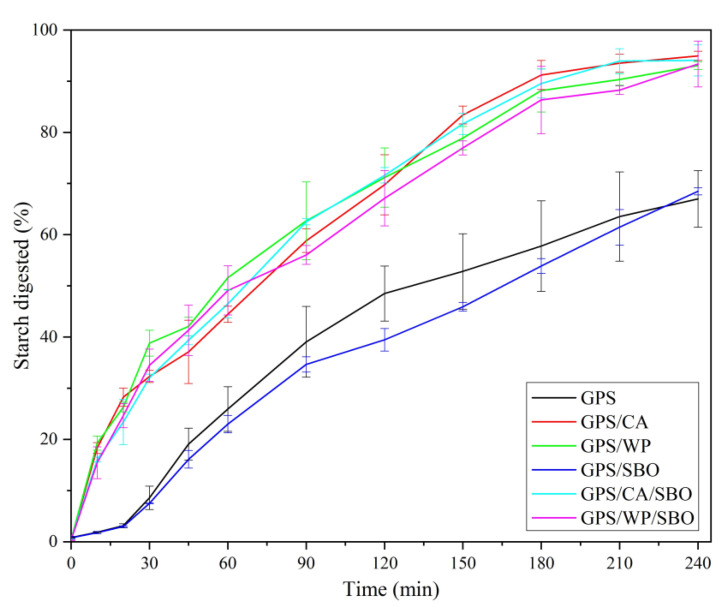
Schemes follow the same formatting. Starch Digestograms of gelatinized potato starch (GPS), gelatinized potato starch/casein (GPS/CA), gelatinized potato starch/whey protein (GPS/WP), gelatinized potato starch/soybean oil (GPS/SBO), gelatinized potato starch/casein/soybean oil (GPS/CA/SBO), and gelatinized potato starch/whey protein/soybean oil (GPS/WP/SBO) during the intestinal phase.

**Figure 3 foods-12-02451-f003:**
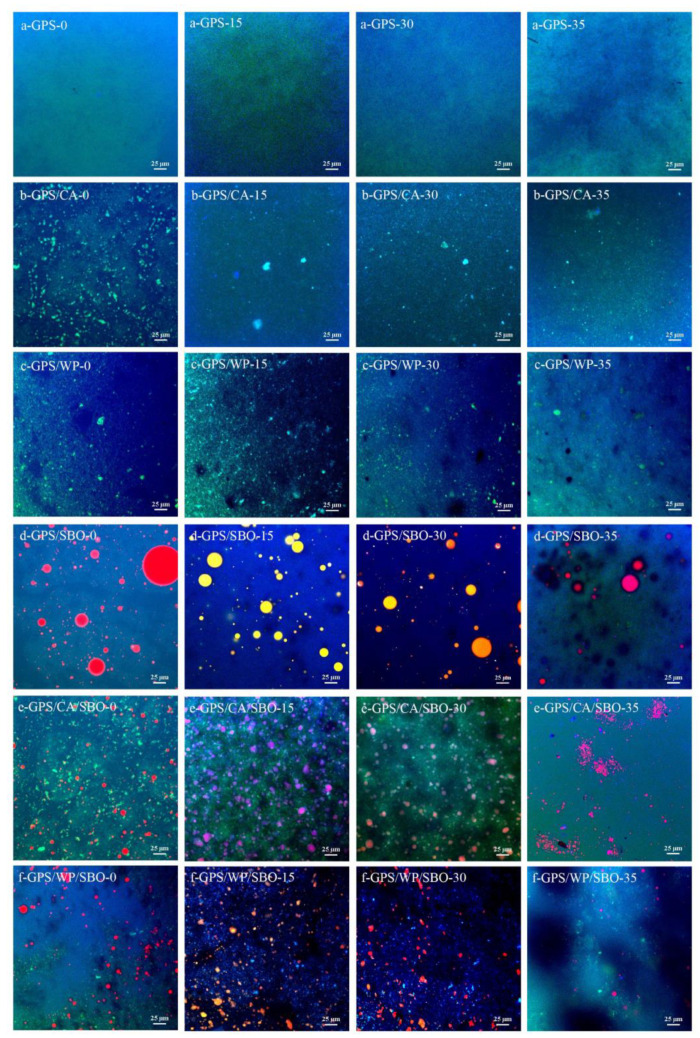
Confocal laser scanning microscopy (CLSM) images of the fresh samples and the samples during the in vitro digestion process (Stomach phase 15 min and 30 min; Small intestine phase 5 min).

**Figure 4 foods-12-02451-f004:**
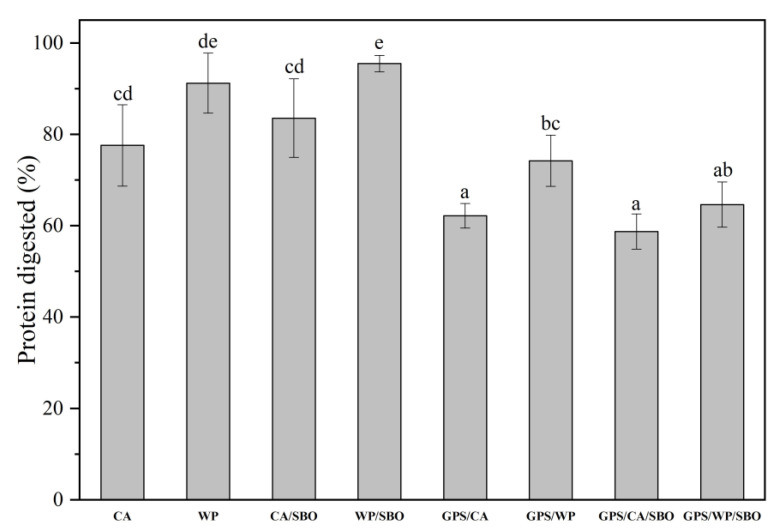
Protein digestibility of Casein solution (CA), Whey protein solution (WP), Casein/soybean oil (CA/SBO), Whey/soybean oil (WP/SBO), GPS/CA, GPS/WP, GPS/CA/SBO, and GPS/WP/SBO. Values with the same letters on graphical bars are not significantly different (*p* > 0.05).

**Figure 5 foods-12-02451-f005:**
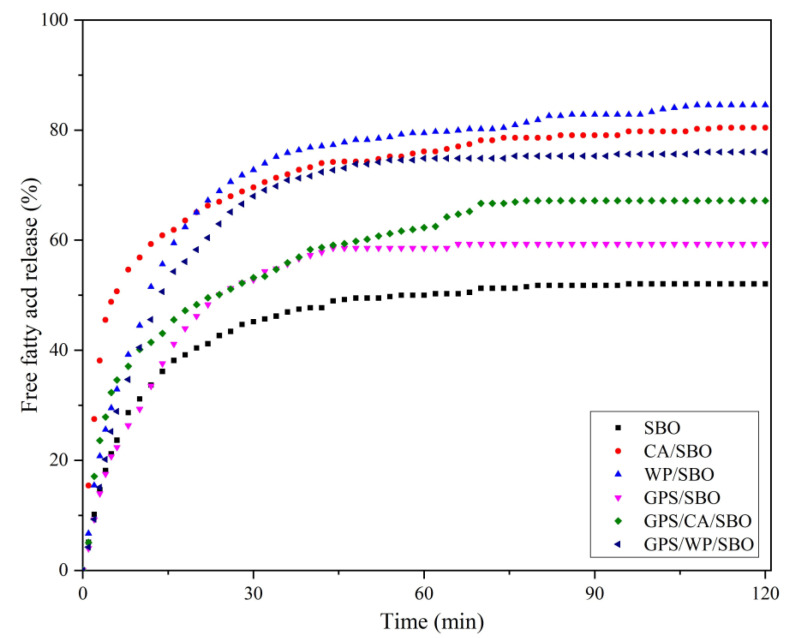
Free fatty acid release profiles from oil-water mixtures (SBO), CA/SBO, WP/SBO, GPS/SBO, GPS/CA/SBO, and GPS/WP/SBO during in vitro intestinal digestion.

**Figure 6 foods-12-02451-f006:**
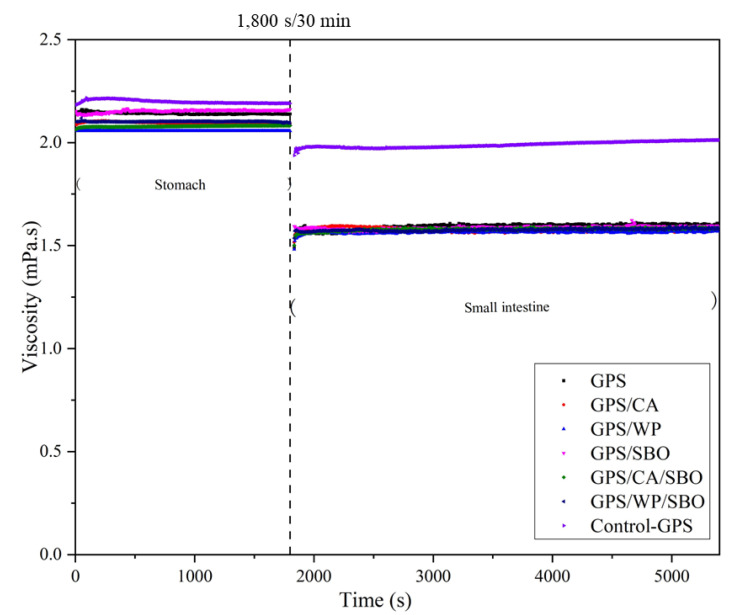
Effect of in-vitro digestion on viscosity of Control-GPS, GPS, GPS/CA, GPS/WP, GPS/SBO, GPS/CA/SBO, and GPS/WP/SBO. Control-GPS did not add the corresponding enzyme during in vitro digestion.

**Table 1 foods-12-02451-t001:** Chemical composition of the raw materials.

Chemical Compositions	Potato Starch	Micellar Casein	Whey Protein Concentrate	Soybean Oil
Moisture (%)	17.90 ± 0.03	7.92 ± 0.08	5.08 ± 0.09	0.06 ± 0.00
Protein (%)	0.13 ± 0.13	81.25 ± 2.86	78.67 ± 4.65	0.00 ± 0.00
Ash (%)	0.23 ± 0.04	7.38 ± 1.45	3.14 ± 0.06	0.05 ± 0.00
Fat (%)	0.22 ± 0.06	0.492 ± 0.04	0.795 ± 0.06	99.9 ± 0.01
Starch (%)	76.77 ± 2.80			
Amylose/Starch (%)	31.14 ± 0.01			

**Table 2 foods-12-02451-t002:** Model parameters, hydrolysis index (HI), and estimated glycaemic index (GI) of the fresh paste samples (in vitro method).

	*D_0_*	*D_∞_*	*k* × 10^−3^	AUC × 10^3^	HI	Estimated H_90_	Estimated GI
GPS	0.82 ± 0.14 ^a^	99.18 ± 0.14 ^a^	4.82 ± 0.97 ^b^	9.85 ±1.32 ^b^	58.07 ± 7.8 ^b^	35.66 ± 5.56 ^b^	70.31 ± 4.47 ^b^
GPS-CA	0.43 ± 0.07 ^b^	99.57 ± 0.07 ^a^	10.98 ± 0.21 ^a^	15.58 ± 0.14 ^a^	91.83 ± 0.80 ^a^	62.98 ± 0.74 ^a^	90.95 ± 0.52 ^a^
GPS-WP	0.40 ± 0.06 ^b^	93.75 ± 0.51 ^c^	13.67 ± 1.85 ^a^	15.95 ± 0.71 ^a^	94.01 ± 4.18 ^a^	66.58 ± 4.26 ^a^	93.02 ± 2.91 ^a^
GPS-SBO	0.79 ± 0.27 ^a^	99.21 ± 0.27 ^a^	3.97 ± 0.18 ^c^	8.65 ± 0.24 ^b^	50.96 ± 1.41 ^b^	30.63 ± 0.94 ^b^	66.26 ± 0.78 ^b^
GPS-CA-SBO	0.42 ± 0.07 ^b^	99.31 ± 0.122 ^a^	10.81 ± 0.27 ^b^	15.47 ± 0.17 ^a^	91.17 ± 1.01 ^a^	62.52 ± 0.73 ^a^	90.58 ± 0.58 ^a^
GPS-WP-SBO	0.31 ±0.06 ^b^	95.47 ± 1.27 ^b^	11.72 ± 0.86 ^ab^	15.37 ± 0.21 ^a^	90.55 ± 1.21 ^a^	62.49 ± 1.82 ^a^	90.39 ± 1.08 ^a^

Values are means ± standard deviations (*n* = 2). Values with the same letters in the same column are not significantly different (*p* > 0.05).

## Data Availability

The data presented in this study are available on request from the corresponding author. The data are not publicly available due to some reasons according to the policy of funding agency.

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
