# Peer review of "The Microstructure, Rheological Characteristics, and Digestibility Properties of Binary or Ternary Mixture Systems of Gelatinized Potato Starch/Milk Protein/Soybean Oil during the In Vitro Digestion Process"

_foods, 2023, doi:10.3390/foods12132451_

Round 1
Reviewer 1 Report
This study seems very interesting. However, it can be improved, especially in terms of rheology results.
In the introduction, relations on the rheology and dıgestibility must be mentioned.
In the results, the rheology results must be discussed in terms of digestibility much more detail.
The similarity report resulted 41% without bibliographia. This must be fixed.

Author Response
Response to the reviewer 1 comments
1. This study seems very interesting. However, it can be improved, especially in terms of rheology results.
Thank you very much for your words.
2. In the introduction, relations on rheology and digestibility must be mentioned.
Thank you very much for your suggestion. We wrote more about the correlation of rheology and digestibility on Line 56 to 62 and Line 71-72.
3. In the results, the rheology results must be discussed in terms of digestibility in much more detail.
Thank you very much for your suggestion. We wrote more about the correlation of rheology and digestibility on Line 247 to 248 and Line 420 to 421 even though we already addressed the correlation between rheological characteristics and digestion on Line 403 to 409.
4. The similarity report resulted 41% without bibliographies. This must be fixed.
Thank you very much for your concern. We used the Turnitin program to check the similarity and we got 30% of similarity.
Even though we have tried to change and rewrite the similar sentences, it is quite difficult to avoid the repletion of similar sentences, especially the sentences in methodology which could not write in the other way otherwise the meaning of the sentences will be changed. However, we are intently concerned about that, the similarities in results and discussion are very small that indicated as a very low plagiarism.
Reviewer 2 Report
This manuscript by Yufang Guan et al. describes some work investigating how combinations of carbohydrate-based, protein-based and fat-based foods affect digestability. In my opinion, the work is interesting and the manuscript is well written. I have only a few minor comments, as listed below:
In Eq. 4, what is 'x', please?
L185: What is CLSM, please?
L196: 'measuring'
L 254: I do not understand how: 'Qualitatively, the digested starch obtained...' Do the authors mean 'quantitatively'.
L279: 'comforted' is the wrong word. I presume that authors meant 'assisted'.
The Figures appear to be out of order. Fig. 4 is mentioned in the text before Figs. 2 and 3.
Fig. 2: Please explain what the various labels (a, ab. bc etc.) indicate.
L407-408: The expression 'The high viscosity of GPS was hydrolyzed...' does not make sense. I presume the authors mean that the sample (GPS) was hydrolysed, resulting in a reduced viscosity.
The English language is generally OK. I found a few minor issues, as listed in 'suggestions for the authors'.
Author Response
Response to the reviewer 2 comments
1. In Eq. 4, what is 'x', please?
Thank you very much for your question. The “x” is the denominator of the differentiate equation which is counted from t0 to t∞. However, we decided to delete equation 4 since there is no such equation in the original reference.
2. L185: What is CLSM, please?
Thank you very much for your question. The CLSM is an abbreviation of Confocal laser scanning microscopy which we already corrected this in the manuscript on Line 185.
3. L196: 'measuring'
Thank you very much for your suggestion. We deleted the word “measuring” in front of “cup” to make the sentence clear.
4. L 254: I do not understand how: 'Qualitatively, the digested starch obtained...' Do the authors mean 'quantitatively'.
Thank you very much for your suggestion. We have already changed the word as you suggested.
5. L279: 'comforted' is the wrong word. I presume that authors meant 'assisted'.
Thank you very much for your suggestion. We have already changed the word as you suggested.
6. The Figures appear to be out of order. Fig. 4 is mentioned in the text before Figs. 2 and 3.
Thank you very much for your notice. We already changed the sequence of figures throughout the manuscript.
7. Fig. 2: Please explain what the various labels (a, ab. bc etc.) indicate.
Thank you very much for your suggestion. We put more details about the labels (a, ab, bc, etc.) at the footnote under the figure 3 as shown on Line 340.
8. L407-408: The expression 'The high viscosity of GPS was hydrolyzed...' does not make sense. I presume the authors mean that the sample (GPS) was hydrolysed, resulting in a reduced viscosity.
Thank you very much for your notice. We have already changed the sentence to “GPS was hydrolyzed by α-amylase and amyloglucosidase to sugar, dextrins and maltooligosaccharides, which possess a lower viscosity than GPS.” which makes the content more clear.
9. Comments on the Quality of English Language
The English language is generally OK. I found a few minor issues, as listed in 'suggestions for the authors'.
Thank you very much. We have already changed all English words following the reviewer’s suggestion and they have been checked again by the native speaker.